# Innate Immunity and Alcohol

**DOI:** 10.3390/jcm8111981

**Published:** 2019-11-14

**Authors:** Shinwan Kany, Andrea Janicova, Borna Relja

**Affiliations:** Experimental Radiology, Department of Radiology and Nuclear Medicine, Otto von Guericke University Magdeburg, 39120 Magdeburg, Germany; s.kany@uke.de (S.K.);

**Keywords:** alcohol, DAMP, PAMP, signaling, sterile inflammation

## Abstract

The innate immunity has evolved during millions of years, and thus, equivalent or comparable components are found in most vertebrates, invertebrates, and even plants. It constitutes the first line of defense against molecules, which are either pathogen-derived or a danger signal themselves, and not seldom both. These molecular patterns are comprised of highly conserved structures, a common trait in innate immunity, and constitute very potent triggers for inflammation mediated via extracellular or intracellular pattern recognition receptors. Human culture is often interweaved with the consumption of alcohol, in both drinking habits, its acute or chronical misuse. Apart from behavioral effects as often observed in intoxicated individuals, alcohol consumption also leads to immunological modulation on the humoral and cellular levels. In the last 20 years, major advances in this field of research have been made in clinical studies, as well as *in vitro* and *in vivo* research. As every physician will experience intoxicated patients, it is important to be aware of the changes that this cohort undergoes. This review will provide a summary of the current knowledge on the influence of alcohol consumption on certain factors of innate immunity after a hit, followed by the current studies that display the effect of alcohol with a description of the model, the mode of alcohol administration, as well as its dose. This will provide a way for the reader to evaluate the findings presented.

## 1. Introduction

Alcohol, as one of the most used drugs worldwide, has played a pivotal role in social events and culture since the beginning of civilization, namely because of its side-effects on the human body. The 2014 National Survey on Drug Use and Health reported that over 87% of people older than 18 years drank alcohol occasionally and over 24% acknowledged to have participated in binge drinking in the past month. The global status report on alcohol and health 2014 by the World Health Organization stated that over 3.3 million deaths worldwide were attributed to causes associated with alcohol consumption. While these deaths are mostly impeded by chronic intake, almost a third of all trauma-related deaths are caused by alcohol consumption [1]. Moreover, up to 50% of trauma patients have positive blood alcohol concentrations [2]. Innate immunity plays a decisive role in controlling and resolving the inflammatory response to tissue damage. As alcohol is a well-known immuno-modulatory drug, it impacts the host response by modulating the innate inflammatory cells in their function. However, since there are numerous inconsistencies regarding its impact, here, we review the influence of alcohol consumption on certain factors of innate immunity.

In general, alcohol consumption is associated with numerous cancer entities, predominantly cancers of the mouth, pharynx, and larynx, but also to cancers of the digestive tract and ovaries [3]. Additionally, the risk for liver disease is concordant to the level of alcohol consumption [4]. However, this is in contrast to certain “health benefits,” which alcohol seems to offer, and which require a critical perspective, as alcohol itself is often the cause of diseases, accidents, and traumatic injuries. A prospective study evaluating mortality in relation to drinking habits of British male doctors revealed that those who drank low amounts of alcohol had the lowest mortality compared to control groups, while those who drank large amounts had the highest rates [5]. In addition, the incidence of cardiovascular diseases has been the lowest for males who drank small amounts of alcohol, whereas non-drinkers showed higher incidence rates [6]. This illustrates how diverse the effects of alcohol on the human body are. Furthermore, there are conflicting modulations of health in relation to the amount (low vs. high) as well as the habit (acute or chronic) of drinking. This hormesis is known for other substances and was first described by Paracelsus. A great number of studies and experience from anyone who ever drank alcohol show that human and animal behavior is modulated by alcohol dose-dependently. In conclusion, this can be inhibitory or stimulatory depending on which organ and which dose is investigated [7]. Homer Simpson once said alcohol “is the cause…. and solution…. to all of problems of life.” Paradoxically, this may have been comic relief, but modern medicine provides him some arguments. Therefore, the bilateral effects or hormesis of certain effects of alcohol on the innate immunity will be displayed in this review.

## 2. Experimental Section

The mentioned data in this review was collected via a systematic literature search conducted using PubMed and Google Scholar. The latter was used to identify any publication not indexed in MEDLINE and covered around 80%–90% of English scientific articles available online [8]. Keywords used were “inflammation”, “innate immunity”, “immune cells”, “cytokine”, “neutrophil”, “sepsis”, “systemic inflammatory response syndrome(SIRS)”, “toll-like receptor (TLR)”, “acute alcohol”, “acute ethanol”, “chronic alcohol”, “chronic ethanol”, and “infection”. All articles were scanned for relevance of content and redundant studies were excluded. There was no limitation on time of publication, however, emphasis was put on more recent work. This review includes in vivo as well as in vitro publications. Furthermore, retrospective and prospective clinical studies were included to link experimental work to observation in patients. The respective study design is mentioned as well as the investigated cells/compartments and limitations to help the reader classify the given information.

## 3. Innate Immunity Factors—Preserved through Time

The innate immunity evolved during millions of years. Thus, equivalent or comparable components are found in most vertebrates, invertebrates, and even plants [9]. It actually constitutes the first line of defense against molecules, which are either pathogen-derived or a danger signal themselves, and not seldom both. So-called pathogen-associated molecular patterns (PAMPs) are comprised of highly conserved structures, a common trait in innate immunity [10], such as unmethylated CpG nucleic acid chains of bacteria, whereas CpG motifs in vertebrates are usually methylated [11]. Other PAMPs includes heat shock proteins, peptidoglycan or fibronectin. Similarly, bacterial endotoxin lipopolysaccharide (LPS), which is often released when bacteria are destroyed by antibiotics, is a very potent trigger for inflammation mediated *via* pattern recognition receptors (PRRs) [12]. These receptors are preserved over the course of evolution and thus, not always very specific. Nevertheless, the addition of binding factors and co-receptors enhances their specificity markedly. For instance, TLR4 itself is not sufficient to work properly. Thus, a complex of MD2–CD14–TLR4 binding LPS after it has been captured by LPS-binding protein is necessary for adequate TLR4 signal transduction [13]. Gram-positive microorganisms do not produce LPS but, rather, other structures, for example, lipoteichoic acid, which induces a similar reaction. Many pathogens including viruses or certain types of bacteria infiltrate cells and, therefore, intracellular mechanisms are obligatory to defend against them [14]. A very prominent group of this type of proteins constitutes the nucleotide-binding oligomerization domain (NOD)-like receptors (NLRs), with a N-terminal caspase activation and recruitment domain, which can bind receptor-interacting protein-2, a protein kinase, that in turn may activate nuclear factor k-light-chain-enhancer of activated B cells (NF-κB) and mitogen-activated protein kinase signaling pathways, inducing a response [15]. The exact process of activating PRRs and their signaling is examined in this review by the example of TLRs. The potential intracellular targets of alcohol are depicted in Figure 1.

### 3.1. Pattern Recognition and Downstream Signaling

Pattern recognition receptors are responsible for binding PAMPs, and thereby inducing an immune response. Furthermore, they identify so-called danger-associated molecular patterns (DAMPs), which are molecules often located intracellularly, and which can act as inducers of inflammation in the absence of pathogens. This form of sterile inflammation is further described below. Different classes of PRRs have been identified, including TLRs, NLRs, or mannose-binding lectin [16]. To date, 10 TLRs have been identified in humans [17]. All of them share a similar structure made up of one extracellular domain built of many repeating leucine segments, thus, bearing the name leucine-rich repeats (LRRs). Human TLRs most commonly have 18–27 LRRs [18]. In general, TLRs are subdivided into subclasses according to their localization and primarily recognizing related PAMPs. TLR1, 2, 4, 5, and 6 are found on the extracellular, while TLR3, 7, 8, and 9 are localized on intracellular compartments (i.e., endosome membrane) [12]. The plasma membrane binds TLRs to recognize mainly microbial membrane structures such as lipids and proteins. TLR2 is known to recognize lipoproteins and its dimerization with TLR1 and TLR6 allows the discrimination between triacyl and diacyl lipopeptides, respectively [12,19]. Further, TLR4 has the capacity to recognize structurally and biochemically discontiguous ligands such as LPS from Gram-negative bacteria, fusion protein of respiratory syncytial virus, or endogenous heat shock proteins [20]. Flagellin, a major component of the locomotor system of flagellated Gram-negative bacteria, is recognized by TLR5 [20]. In contrast, intracellular TLRs detect nucleic acids derived from bacteria and viruses and endogenous nucleic acids in the pathogenic context [12]. Here, TLR3, the only TLR that acts MyD88-indepedently, recognizes viral double-stranded RNAs, whereas single-stranded RNAs are detected by TLR7/8. Additionally, TLR7/8 mediates the recognition of small purine analog compounds [12]. Finally, TLR9 senses bacterial and viral DNA that is rich in unmethylated CpG motifs [19]. Functionally, TLR1, 2, 4, 5, and 6 mostly induce production of pro-inflammatory cytokines such as tumor necrosis factor alpha (TNFα), interleukin (IL)-1, IL-6, and IL-10, while TLR7 and 9 primarily induce production of type 1 interferons [14].

Alcohol consumption exerts distinct effects on PRRs, primarily on TLRs. As consequence, either a pro- or anti-inflammatory influence of alcohol consumption on immunity is observed, which actually depends on the dose and timing or duration of exposure to alcohol. After feeding mice with a single dose of alcohol (5 g/kg), TLR4 gene expression is reduced [21]. Not only TLR expression-reducing effects are observed, but acute alcohol consumption also induces a hyposensitivity of TLR4 to ligands such as LPS in rat intestinal epithelial as well [22]. Consequently, in a sepsis mouse model with binge drinking, TLR4-mediated cytokine production is reduced [23]. However, this is time-dependent as the mRNA expression of TLR4 and TLR2 seems to be enhanced in short periods up to one hour but subsequently suppressed after a longer time (e.g., three hours in microglia cells from either TLR2 or TLR4 knockout mice) [24]. Correspondingly, acute alcohol consumption inhibits TLR2-, TLR4-, and TLR9-induced inflammatory response mediated by p38 and ERK1/2 in macrophages, after the mice were given a single intraperitoneal (i.p.) injection of 2.9 g/kg alcohol and subsequently sacrificed after three hours [25]. Moreover, acute alcohol (25 mM) application to human blood monocytes down-regulates TLR8- and TLR4-mediated TNFα protein and mRNA production, while it augments the production of the anti-inflammatory cytokine IL-10 in monocytes [26]. The same group using a similar model of human monocytes under acute alcohol influence has shown that alcohol attenuates TLR4- but not TLR2-induced NF-κB activation, as well as TNFα mRNA and protein production [27]. Further assays revealed the influence of alcohol on TLR4-induced inflammation to be triggered via inhibition of IL-1 receptor-associated kinase (IRAK)-1 and ERK1/2 kinases, as well as an increase in IRAK-monocyte levels. However, in the case of both TLR4 and TLR2 activation, acute alcohol consumption activates IRAK-1 and c-Jun N-terminal kinase phosphorylation, thus inducing an inflammatory response [27].

In contrast to its immune-suppressive impact in the acute setting, chronic alcohol consumption results in upregulated TNFα protein and mRNA expression in human monocytes without significant effects on IL-10 production, effects that were observed regardless of the application of either TLR4 or TLR8 ligands [28]. Interestingly, exposing mice for three or four weeks to alcohol produces higher levels of TLR4 in liver macrophages compared to control mice. Mice fed with alcohol (average blood level 139.1 mg/dl) for ten days had higher expressions of mRNA of all TLRs in the liver, except TLR3 and TLR5, while TLR10 and TLR11 were not tested [29]. The same group shows a higher sensitivity of TLRs to congruent ligands, which has been reflected in increased TNFα levels. While antibiotics do not prevent the induction of TLR mRNA production, inhibition of the nicotinamide adenine dinucleotide phosphate oxidase (NADPH oxidase) is effective in limiting hepatic TNFα levels [29]. These data indicate that reactive oxygen species (ROS) play an important role in inflammation, which is induced by chronic alcohol consumption [29]. Nevertheless, TLR3 examined in a binge-drinking mouse model with TLR3^-/-^ and IL-10^-/-^ knockout mice seemed to have an antagonistic effect to TLR4. Treatment with polyinosinic-polycytidylic acid, a TLR3 ligand, decreases TNFα, IL-6, MCP-1, and FAS gene expression, and enhances IL-10 gene expression in the qRT-PCR analysis in isolated Kupffer cells as well as in hepatic stellate cells [30].

In conclusion, alcohol in its acute use is a potent anti-inflammatory agent and ameliorates the TLR4-mediated pro-inflammatory cytokine response. In contrast, chronic alcohol consumption increases the sensitivity of TLR, subsequently leading to the higher expression of proinflammatory cytokines (e.g., TNFα).

### 3.2. NF-κB—Key to Inflammation

Many downstream signaling pathways end in the activation of the transcription factor NF-κB, with subsequent modulation of its target gene expression. One example is the MyD88 pathway, which begins with MyD88 and its Toll/IL1-receptor (TIR) domain binding to the TIR domain of TLR4 [31]. MyD88 also has a death domain, which is used to bind IL-1 receptor-associated kinases (IRAK) 1 and 4 [32]. This IRAK-1–IRAK-4 complex recruits TNFα receptor-associated factor (TRAF) 6, hereby enabling TRAF6 to enlist serine/threonine kinase TGFβ-activated kinase (TAK) 1 and TAK-binding proteins TAB 1 and 2 [33]. TAK is activated by this process via phosphorylation [34]. Now, converging different pathways, the central protein is made up of three subunits; two catalytic subunits, IKKα and IKKβ, and a regulatory subunit, IKKγ (also named κ-κB essential modulator or NEMO) [35]. Then, an inhibitor of NF-κB kinase (IkB)-complex (IKK) is activated by phosphorylation [35]. IKK phosphorylates IkB, thus exposing it to disassembly via ubiquitination [36]. After the dissolution of IkB, it can no longer inhibit NF-κB. This transcription factor, originally composed of p50, a processing product of p105, and p65 (also called RelA), can translocate to the nucleus [37]. Finally, NF-κB binds to enhancer elements to produce cytokines and adhesion molecules to accelerate inflammation and diapedesis of the effector cells of innate immunity, neutrophils, and macrophages for example [38].

Another pathway to activate NF-κB is non-canonical signaling, which relies on the tightly-regulated processing of p100, opposed to the rather constant processing of p105 [39]. The canonical NF-κB pathway is responsive to numerous different receptors such as TLR4, IL-1, TNFR, and T-cell receptors [40]. This is in contrast to the non-canonical pathway, which is mostly activated by receptors from the TNFα receptor superfamily [41], including activator of nuclear factor kB [42]. In an unstimulated milieu, the above mentioned p100 processing is inhibited by degradation of NF-κB-inducing kinase (NIK) [43]. Here, TRAF3 rapidly binds the newly synthetized NIK and induces its ubiquitylation by recruiting of E3 ligases cellular inhibitor of apoptosis (cIAP), needing TRAF2 as an adaptor molecule [44]. Upon activation, the TRAF2–TRAF3–cIAP complex is recruited to the TNFα receptors and its subsequent ubiquitylation and degradation lead to NIK accumulation [44,45]. IKKα is activated by this accumulation independently on trimerization with IKKβ and IKKγ, in contrast to the canonical pathway [46]. Active IKKα ensures the processing of p100 and is followed by translocation of p52–RelB heterodimer into the nucleus to finally modulate NF-κB gene expression [47]. Additionally, IKKα phosphorylates NIK and is thereby regulating itself via a negative feedback loop [47].

Although NF-κB signaling is detrimental in the alcohol-induced innate immune response, there is still a knowledge gap on the level of its target genes, caused probably by the individual variations of NF-κB activation. NF-κB is expressed at high levels in microglia and other monocyte-like cells among low levels of innate immune genes in homeostasis. Upon ethanol administration, the NF-κB–DNA binding increases and the transcription of various target genes is induced, including chemokines (CCL2), pro-inflammatory cytokines (TNFα, IL-1β, IL-6), and pro-inflammatory oxidases (NOX, COX, iNOS) or proteases (TACE, tPA) [48,49]. However, this seems to be dose-dependent, since leukocytes of moderate alcohol-drinking individuals exhibit lowered NF-κB levels in acute and chronic settings [50]. Moreover, significant dysregulation of genes critically involved in wound healing, blood coagulation, cancer, cardiovascular, and lung diseases was shown in chronic heavy drinkers [51,52].

Experimental settings have shown that Kupffer cells isolated from Sprague-Dawley rats, which were stimulated with 100 mM alcohol for 90 min, yielded in inhibition of LPS-induced NF-κB activation, which was demonstrated by depressed levels of TNFα mRNA and TNFα secretion [53]. The exact target of alcohol remains contested, as monocytes treated acutely with alcohol have shown increased levels of p50 translocation to the nucleus, leading to an increase of inhibitory p50–p50 complex, and a successive decrease of cytokine levels, while p65/RelA remained unaffected [54,55]. Conflicting with the findings, exposure of human melanoma cells (FEMX-I cells) to alcohol (100–400 mM) for 24 hours ensued in a concentration- and time-dependent increase of NF-κB activation, up to 900%, as compared to control cells [56]. Interestingly, this is associated with a suppression of p50–p50 homodimers that normally block transcription [56]. During acute incubation of human monocytes or murine macrophages with alcohol, NF-κB activity has been inhibited independent of IkappaB alpha degradation, indicating another target molecule that may be involved in NF-κB suppression by acute alcohol exposure [57,58]. Another group has provided more data on alternative ways of alcohol NF-κB modulation; rats were exposed to 6 g/kg body weight alcohol (by oral gavage with a 32% v/v solution), then injected with LPS (60 ug/mouse i.v.) after 30 min. Analyses of peritoneal macrophages followed 15, 30, 45, 60, and 75 min after injection [59]. While alcohol does not influence the LPS-induced p65 translocation, transgenic mice with an NF-κB luciferase reporter show that NF-κB is significantly inhibited by alcohol. However, exact information on how alcohol led to this effect is not evident in this experiment. The applied doses of alcohol (4 or 5 g/kg) yielded blood levels of 43.5 and 65.2 mM, respectively [59]. Murine alveolar type 2 epithelial cells exposed to alcohol (0–100 mM), followed by LPS and TNFα exposure, after 90 min displayed potent and dose-independent inhibition of CXCL5 expression and impairing of p65 phosphorylation [60]. Mice fed with an alcohol-containing diet for four weeks demonstrated increased levels of NF-κB phosphorylation, steering higher activation of NF-κB [61,62]. Acute alcohol consumption amplifies the IL-1R-associated kinase monocyte (IRAK-M), a suppressing mediator of IRAK-1 in human monocytes [63]. This is linked to depressed IkappaB alpha kinase activity, NF-κB–DNA binding, and NF-κB driven cellular responses [63]. In contrast, chronic alcohol reduces IRAK-M activity but augments IRAK-1 and IKK kinase levels, as well as NF-κB–DNA binding and responses [63]. In conclusion, Mandrekar et al. show that acute alcohol use leads to a suppressed monocyte reaction to LPS, whereas chronic administration amplifies the monocytic response to LPS. Female mice fed with alcohol (5 g/kg) prior to LPS injection one or 24 hours later exerted decreased levels of IRAK-1 and LPS-mediated activation of NF-κB after one hour in Kupffer cells, but, in contrast to this, yielded to augmented levels of IRAK1 and LPS-induced NF-κB activity after 24 hours [64]. Male mice fed a Lieber-DeCarli diet containing EtOH or an isocaloric control (ctrl) diet for four weeks have shown increased systemic pro-inflammatory cytokine levels, and enhanced local gene expression of NF-κB-controlled genes, such as intercellular adhesion molecule (ICAM)-1 and matrix metallopeptidase (MMP)-9, as well as enhanced c-Jun and NF-κB phosphorylation after massive blood loss [62]. Therefore, a certain hormesis of alcohol is evident with regard to this pathway, as its induction or inhibition clearly appear correlated to dose and duration of the exposure to alcohol.

## 4. Pathogen-Associated and Sterile Inflammation

Immunity does not only differentiate between self and not self but also between dangerous and not dangerous. This danger hypothesis postulated by Polly Matzinger explains why the human body can induce so-called sterile inflammation [65]. Here, danger molecules are intracellular proteins or nucleic acids, which are not found outside the cell compartment under normal conditions [66]. However, upon cellular damage the danger molecules are released into plasma or extracellular space, thus providing signals to induce inflammatory processes via PRRs by binding of those molecules [67]. To mention only a few, danger-associated molecules include high-mobility group box 1 (HMBG1), adenosine triphosphate (ATP), uric acid, deoxyribonucleic acid (DNA), or degraded extracellular matrix like heparan sulfate and hyaluronan [68]. The intracellular PRR class includes the RIG-I-like receptor, the AIM2-like receptor (ALR), and NLR proteins, whereby the latter two assemble cytosolic complexes called inflammasomes to activate pro-inflammatory caspases-1 and -11 in a canonical and non-canonical manner, respectively [69]. Most inflammasomes use an adaptor protein “apoptosis-associated speck-like protein containing a caspase activation and recruitment domain”, so called ASK, which recruits monomers of pro-caspase-1 to close spatial proximity, leading to proteolytical self-cleavage and formation of active caspase-1 tetramers. Active caspase-1 induces pyroptosis and proteolytically cleaves pro-IL-1β and pro-IL-18 to their active forms, followed by release from the cell that causes activation and recruitment of other immune cells [69,70].

The concept of sterile inflammation is shown in this review, as an example, on the chromatin-associated protein HMGB1, which normally binds to DNA and functions as a chaperone [71]. HMGB1 is released from the nucleus and the cell upon stimuli like IFNɣ or TNFα [72], or from apoptotic cells, [73] among others. In extracellular space, HMGB1 can induce the inflammatory response via TLR4 and RAGE [74]. Furthermore, it can be actively released from immune cells such as macrophages [75]. This underlines the importance of messenger molecules; whose release do not necessarily depend on pathogens. Another rather prominent member of this group is ATP, which is known for its extensive role in the cell metabolism. Outside the cell, ATP has a fairly different role from its original characteristic, now acting as a signal for cell damage via activating purinergic (P2) receptors [76]. Activation of purinergic receptors by ATP or other nucleotides has potent autocrine and paracrine effects on cellular functions [77,78], whereby the receptor density particularly contributes to the outcome of purinergic receptor-mediated signaling [76]. For example, via P2X receptors, ATP induces production of cytokines such as of the pro-inflammatory IL-1β and IL-2, for example, among many others [79,80].

The resistance to pathogen-associated and sterile inflammation via DAMPs is modulated by alcohol, which is best evident in chronic alcohol abusers as well as in conditions of acute binge drinking. Mice treated with 6 g/kg alcohol, resulting in a blood concentration of 87 mM, were injected with LPS from *Escherichia. coli*. After only two hours, alcohol-treated mice displayed a suppressed bacteria clearance and lowered production of pro-inflammatory cytokines compared to the control group [23]. Rats infused intraperitoneally with a single bolus of alcohol (5.5 g/kg) were administered *Streptococcus. pneumoniae* intratracheally and studied for up to 40 hours. Total bacteria numbers in the lungs were similar between controls and the alcohol group at six hours after injection. However, after 18 hours, *S. pneumoniae* levels significantly decrease in the control group compared to the alcohol group [81]. Furthermore, alcohol-exposed rats are more likely to exhibit a systemic dissemination of bacteria to spleen, blood, and liver 40 hours after administration of *S. pneumoniae* [81]. Not only the clearance of bacteria is impaired, but most prominently the gut barrier is weakened as well [82]. Acute alcohol binging (2 mL of vodka 40% v/v alcohol/kg body weight) leads to a rapid rise in serum endotoxin and 16S ribosomal DNA levels in human volunteers [83]. Interestingly, compared to men (*n* = 11), woman (*n* = 14) have strongly elevated blood alcohol levels and serum endotoxin levels [83]. Hence, gender-based differences in the metabolization and influence of alcohol on immunity must be assumed and have to be considered in animal models as well. Alcohol users, which were reviewed over a three-year period, reveal a fatal weakness to bacterial infection, most eminent in a group that is infected with bacteremic *Klebsiella pneumoniae*. Actually, they show a mortality rate of 100%, a short onset of illness before hospital admission, and a short survival after admission (24.6 ± 7.9 h). It must be pointed out that the examined group was rather small (*n* = 11) and included heavy smokers and comorbidities like diabetes or liver cirrhosis [84]. In another study with 1161 burn patients, those with high blood alcohol concentration (BAC) (>0.1 g/100 mL) are more susceptible to infectious disease and pneumonia compared to the control population [85]. Furthermore, as reported recently, the constitutive microbiome of the gut can be altered by chronic alcohol consumption as well [82]. Mice fed with an alcohol-containing diet for eight weeks show a differential composition of the gut biome as compared to the control group [86]. In the alcohol group, a decline of Bacteroides and *Firmicutes phyla* is observed, while a rise of *Alcaligenes* and *Corynebacterium*, as well as Actinobacteria phyla, can be determined [86]. A chronic model with male rats gavaged for 10 weeks with 6 g/kg/day alcohol uncovers a significant gut leakiness (determined via oral sugar test) after two weeks of alcohol consumption, which is followed by an endotoxemia after only four weeks [87]. The authors suspect that oxidative stress may be a contributing factor to leakage because levels of the inducible nitric oxide synthase (iNOS) in gut mucosa were increased to 400% as compared to controls [87]. Yet, the exact cause of augmented LPS translocation or rather gut “leakage” remains to be determined; however, cytokines, zinc, and hormones constitute the prime suspects [88]. The Kovacs group established a model of combined alcohol insult and burn injury. Here, mice are injected a single bolus of alcohol (1.12 g/kg) and placed in a plastic template that exposes 15% of total body surface, followed by a 95 °C bath that leads to a full-thickness burn injury [89]. Early studies of this model reveal an impaired intestinal barrier as well as an overgrowth of bacteria, which leads to a translocation of pathogens into the bloodstream after alcohol application [90]. Additionally, in human neuroblastoma cells, which were treated with 50 mM alcohol, the HMGB1 expression and release are elevated after 12 and 24 hours [91]. In the same cell culture, lactate dehydrogenase, as an indicator of cell damage, is not increased compared to the control. Thus, suggesting that HMGB1 does not necessarily derive from necrotic but rather from intact cells [91]. This provides a new perspective of otherwise “healthy” tissue contributing to systemic inflammation. Another important target for alcohol is purinergic receptors, which were briefly addressed above in ATP signaling. Male C57BC/6J mice administered with a bolus of alcohol (3.5–5.5 g/kg) express an increase of P2X7R in alcohol-sensitive brain regions [92]. Nonetheless, the effects are specific to certain receptors of the P2X family [93]. In *Xenopus* oocytes, alcohol potentiates ATP-gated P2X3 receptors to increased activity, yet, it inhibits P2X4 receptor functions [93]. Interestingly, recent research indicates that P2X3 receptors potentially play an important role in chronic pain that alcohol may worsen [94].

In conclusion, alcohol can enable pathogens to enter the systemic blood flow, a process that may lead to an increased susceptibility of patients with infections. Furthermore, the induction of DAMPs in a sterile environment by alcohol should be a focus of further research, because, potentially, this may provide novel understanding of the chronic inflammation after alcohol consumption in case of no visible damage to organs.

## 5. Cytokines

Cytokines are small proteins (<40 kDa), which are produced and secreted by almost every cell in response to DAMPs or PAMPs, and are important mediators of cell-to-cell communication to regulate and influence immune responses. The release of proinflammatory cytokines can lead to the activation of other immune cells and production as well as the release of further cytokines to amplify or dampen the immune reaction. Consequently, the term “cytokine storm” arose, which characterizes inflammation as a sudden release of cytokines to (up-) regulate inflammation. Depending on its signaling, IL-6 has pro-inflammatory and anti-inflammatory properties, and plays the central role in activating and maintaining inflammatory responses [95,96]. It can modulate the recruitment of innate immune cells via influencing other chemokines, including CXCL1 and CXCL8 (IL-8), monocyte-attracting chemokines CCL2/MCP-1 and CCL8/MCP-2 [97], or cell adhesion molecules such as vascular cell adhesion molecule (VCAM), ICAM, and E-selectin [98]. Importantly, despite its prevalent pro-inflammatory character, in inflammatory disease models and knockout mice models, IL-6 is essential for adequate liver regeneration, gut barrier repair, and suppression of inflammation [99,100,101]. Another major pro-inflammatory cytokine is IL-1, including IL-1α and IL-1β, which was the first to be discovered in the early 1970s by Charles A. Dinarello, and since then has been greatly studied [102]. Secretion of IL-1α protein is well regulated as it can exert properties of an alarmin [103]. In contrast to IL-1α, the IL-1β precursor is not biologically active, as its functionality requires a proteolytic cleaving by the IL-1β-converting enzyme caspase-1 within the multiprotein inflammasome complex [104]. The primary source of IL-1β are monocytes and macrophages, like microglia or Kupffer cells as well as dendritic cells [105]. Models with IL-1-deficient mice show that those are rather prone to bacterial, mycotic, and protozoa infections, as compared to their wild type controls [105,106]. Moreover, exaggerated IL-1β processing is fundamental for the development of autoinflammatory diseases [107]. It has been shown that type 2 diabetes patients exhibit not only elevated IL-1β levels, but also a decrease of its antagonist IL-1Ra in pancreatic islets, suggesting an imbalance causing excessive inflammation and failure of the beta cells to secrete a sufficient amount of insulin [107]. Tumor necrosis factor alpha was first described in 1975 by Carswell et al. for its cytotoxic activity via immune cells and has; thus, gained its name TNF [108]. The main source of TNFα are macrophages and T-cells, yet many other cells such as B-cells, neutrophils, and endothelial have been reported to produce it as well [109]. TNFα plays an important role in cell survival and the pro-inflammatory response via NF-κB and AP-1 [110], and Fas or caspases as well [111,112]. While on the one hand, TNFα knockout mice are protected against shock, they are far more susceptible to bacterial challenge [113]. Nonetheless, TNFα antagonists are highly effective for the treatment of auto-inflammatory diseases like psoriasis, Crohn’s, or rheumatoid arthritis [110]. Interestingly, those patients treated with a TNFα antagonist displayed a reactivation of latent tuberculosis (TB) or even an aggravation of primary TB [114]. It has been shown that TNFα is crucial for the macrophage activation and recruitment with subsequent formation of granulomas in order to limit the infection of *Mycobacterium tuberculosis*. Despite these anti-mycobacterial features, TNFα directly promotes *M. tuberculosis* growth in human monocytes, and its cytotoxic character contributes to tissue damage and necrosis of TB lesions that in turn may contribute to organ dysfunction [114]. This shows clearly the dual role of TNFα, but the pathological mechanisms still have to be investigated. In 1989, IL-10 was first described by Fiorentino et al. as a cytokine synthesis inhibitory factor [115]. It is produced by almost all leukocytes including macrophages, dendritic cells, neutrophils, NK cells, B-cells, and CD8^+^ T-cells, whereby CD4^+^ T-cells are the major producers [116,117]. IL-10 signaling in a subpopulation of those, in so-called regulatory T-cells (Tregs), has been shown essential in selective regulation of pathogenic Th17 cell responses [118]. The biological effects of IL-10 on innate immune cells suppress the release of immune mediators, antigen expression, and phagocytosis [119]. IL-10 prevents polymorphonuclear leukocytes (PMNs) activation and TNFα, as well as IL-8 release upon LPS administration [120], but it also attenuates TNFα-induced ROS production, ICAM-1 expression, and leukocyte adhesion to human umbilical vein epithelial cells [121]. IL-8 was initially observed for its trait as a chemoattractant for granulocytes, primarily neutrophils in vitro [122]. NF-κB and JNK, as well as AP-1, have been identified as the main pathways for inducible IL-8 expression [123]. Each cell that is expressing TLR can produce and secrete IL-8, including macrophages and smooth muscle cells [124], while endothelial cells accumulate it in vesicles known as Weibel–Palade bodies [125]. Interleukin-8 is decisive for the process of neutrophil homing and transmigration, but it can influence the respiratory burst as well [126]. In conclusion, IL-8 is a very potent trigger to cell migration and proliferation and thus should always be considered in inflammation models.

Cytokines are affected by alcohol on several levels as they are induced by certain pathways affected by alcohol, which again, in turn, can be modulated by other cytokines. Summarized, this makes it difficult to differentiate between altered cytokine actions and altered cytokine release. In an in vitro model of acute inflammation, pretreatment of human lung epithelial cells with alcohol (85 or 170 mM) for 24 or 72 hours reduces IL-8 release upon their stimulation with IL-6. In contrast to the treatment of cells prior to inflammatory stimulation, treating cells with alcohol afterward reduces the IL-8 release significantly after an incubation period of one hour. Similar findings are reported in vitro for the IL-6 release [127,128]. In vivo findings of blood samples obtained from mice, which were exposed to alcohol (6 g/kg) before being injected with 2 × log ² *E. coli* (intraperitoneally), display a decrease in the systemic CXCL9 release, increased IL-10, and lowered IL-6 and IL-12 production after 17 hours. These data underline not only the reduction of pro-inflammatory interleukins but an increase of anti-inflammatory cytokines in serum samples as well [129]. In line with these results, using the same binge drinking model, wild-type mice show decreased levels of IL-15, TNFα, IL-9, IL-1β & IL-1α, IL-13, IL-17, and IL-6, while IL-10 and MIP-2 are increased in the peritoneal lavage fluid [23]. However, this is not represented in each compartment of the body, as acute alcohol use may deter TNFα production in serum, but, on the other hand, bronchoalveolar lavage fluid TNFα levels in the mouse model were not altered at any time after infection [81]. Importantly, it adds another dimension to alcohol’s modulation of immunity, because the observed effects may be exclusive to the investigated location. This spatial aspect has to be considered when discussing the subject. In human monocytes treated with 25 mM alcohol, short term exposure (one to two days) reduces the LPS-induced TNFα release and gene expression. Interestingly, prolonged exposure (four to seven days) increases TNFα production in human monocytes upon LPS treatment, indicating that acute and chronic alcohol exert different effects [63]. This is supported by other mouse models of chronic alcohol consumption, showing that chronic use enhances the LPS-induced hepatic mRNA expression of TNFα, IL-6, and IL-10 [130]. Similar data are provided in human pathologies, which are caused by excessive or chronic alcohol consumption. Serum levels of pro-inflammatory cytokines TNFα, IL-6, and IL-1 are increased in patients with advanced alcoholic liver disease [131]. In patients admitted to hospital with acute alcoholic hepatitis, serum levels of IL-8, IL-4, and IFNγ are higher than age- and sex-matched control patients [132]. The severity of this alcohol-induced hepatitis directly correlates with cytokine concentration, yet they can normalize after recovery [133].

Regarding the above-discussed studies with aforementioned diseases, the confounding factors have to be considered as well. Therefore, analyses of cytokine alterations by alcohol exposure in healthy individuals are imperative to understanding alcohol’s influence on the innate immunity. Such work has been performed by the Kovacs group. Blood samples from intoxicated healthy volunteers (*n* = 15, blood levels 0.81–0.89 g alcohol/kg body weight) were taken from 20 min until five hours after alcohol consumption and were stimulated with LPS [134]. A temporary pro-inflammatory state with enhanced LPS-induced TNFα expression after 20 min, followed by an anti-inflammatory state with attenuated LPS-induced IL-1β levels from two until five hours is observed after binge drinking [134]. Another comparable study with healthy male subjects (*n* = 20, 4.28 ml vodka/kg body weight) reports increased IL-1Ra serum levels within two hours after alcohol consumption [135]. This indicates that the modulation of the immune system by alcohol cannot be simplified as pro- or anti-inflammatory, but rather as periodic. A prospective cohort study with 8209 subjects (69% men, median 50 years) supports the data from the point of view of the moderate drinkers [136]. Drinking habits were measured for ten years between 1984 and 1995, and analyses were carried out three times during the following 12 years. Stable non-drinkers, as well as stable heavy drinkers, show higher levels of circulating IL-6, C-reactive protein (CRP), and IL-1Ra compared to moderate drinkers [136]. This survey is remarkable because the authors have established a longitudinal approach in determining drinking habits over ten years and not only the habit at one point in time. However, the population is very male dominated, and the drinking habits have been evaluated by a self-report from the subjects and were not analyzed objectively. Other studies indicating the anti-inflammatory potential of an acute alcohol consumption have been performed in the cohort of traumatized patients. As noted in the introduction to this review, admission of trauma patients to emergency departments is often in relation to alcohol consumption. Severely injured patients with positive blood alcohol concentration (BAC ≥ 0.5‰, *n* = 49) exhibit lower circulating IL-6 levels and leukocyte counts compared to non-intoxicated trauma patients [137]. Similar results for intoxicated patients with traumatic brain injury have been reported as well [138]. To further examine cytokine modulation by alcohol in case of trauma, an established murine model of alcohol and burn injury has been applied, which we have previously addressed in this review. Using this model, mice display increased systemic levels of IL-6 and TNFα and raised levels of IL-6 in lungs [139,140]. This model has also been extensively used to investigate the role of interleukins in intestinal damage. Indeed, mice treated with alcohol before burn injury show exacerbated levels of IL-6, IL-1β and IL-18 protein in ileal tissue [141,142]. While treatment with IL-6 antibody does not significantly decrease IL-6 levels in serum or ileum, a reduction of morphological mucosa damage as well as bacterial translocation can be observed [143]. Recent studies indicate that alcohol can suppress the inflammasome activation and thereby influence pro-inflammatory cytokine release [144]. Human macrophages of the THP1 type were exposed to LPS (1 ug/mL, three hours) and treated with activators of the NLRP3 inflammasome like ATP, cholesterol crystals, and others, and with alcohol as well (43, 86, 171, and 343 mM) [145]. The data show that alcohol suppresses the release of IL-1β dose-dependently [145]. Nevertheless, alcohol did not affect the expression of NLRP3 or IL-1β mRNA in macrophages, but the data correlated caspase-1 activation with suppressed IL-1β release [145]. The authors came to a conclusion that alcohol inhibits the inflammasome activation instead of directly influencing IL-1β. On the other hand, in a sepsis model, male Wistar rats were given 10% alcohol solution as the only source of liquid for four weeks (leading to a median of 11.7 g/kg alcohol intake). After that, they were exposed to an i.p. fecal injection to induce sepsis. Rats with higher levels of consumed alcohol (>11.7 g/kg) have higher mortality rates in early sepsis [146]. Yet, the levels of IL-6 and TNFα in serum are attenuated by alcohol in case of the sepsis group compared to the control sepsis group [146]. In a model of escalating alcohol consumption, rats have been subjected to lavage of gradually increasing alcohol concentration (5 to 9 g/kg body weight) for 16 weeks. This increases the levels of IL-1, IL-10, and TNFα in serum [147]. Interestingly, IL-10 is upregulated between weeks four and eight but declines significantly thereafter [147]. This infers to a progressively pro-inflammatory milieu after chronic intake of alcohol and supports the idea of periodic immunomodulation. Interestingly, the effects of alcohol appear to differ depending on the second hit or the trigger mechanism. In a rat model of chronic alcohol exposure, alcohol alone did not alter plasma levels of TNFα, IL-10, IL-6, or CXCL1. Yet, in the presence of LPS, alcohol doubled the inflammatory response with strong increases of IL-6 and TNFα levels [148]. Another chronic alcohol consumption model shows a significant rise in systemic levels of IL-6, MCP-1, and TNFα in alcohol-fed mice compared to controls [61]. Furthermore, the inflammatory potential of Kupffer cells after LPS stimulation has been examined using their supernatants. The data reveal the highest concentration of TNFα after only two hours, while IL-6 had the highest concentration at 24 hours after consumption [61]. Certainly, this could imply TNFα to be the primary response after LPS and alcohol consumption, which may be followed by a more dilatory IL-6 rise in Kupffer cells. Not only the liver but also the lungs of patients with alcohol use disorder (AUD) are under great pro-inflammatory stress, as evident in the examination of those patients’ bronchoalveolar lavage (BAL) (*n* = 19, control *n* = 20) [149]. Human alveolar macrophages isolated from those BAL have a significant rise of TNFα, IL-8, CXCL10, and CCL5 protein levels. In addition, IL-1β, IL-6, and IL-1Ra proteins are enhanced as well [149]. Furthermore, as mentioned before, there seems to be a different response regarding gender. A chronic alcohol model using rhesus macaques uncovers higher expression of IL-1β, IL-15, and IL-8 in females compared to males [150].

In conclusion, the evidence for alcohol to greatly influence cytokine production is indisputable. Further clinical studies using healthy subjects will point to certain cytokines that may be usable as biomarkers for alcohol disease or for its immuno-modulatory impact. 

## 6. Cellular Responses

### 6.1. Cellular Responses—Phagocytosis and Oxidative Burst

The first cells to respond to pathogens are usually those that also have the ability to directly and independently neutralize and kill the microbes by, for example, phagocytosis or ROS. This is the process of recognizing pathogens and swallowing them to digest and destroy [151]. The main populations of phagocytic cells are composed of monocytes and macrophages, neutrophil granulocytes, and dendritic cells, yet even epithelial, Sertoli cells, or retinal cells provide phagocytosis [151]. Potential target points for alcohol in inflammatory tissue are shown in Figure 2.

Monocytes originate from myeloid precursor cells in fetal liver and bone marrow in adult and embryonic hematopoiesis [152]. There are two main populations in humans. In brief, the first population constitutes CD14^+^ cells that are either CD16^+^ (a receptor for Fcγ of immunoglobulins) or CD16^–^ [153]. The second population is made up of CD14^low^/CD16^+^ cells [154]. While these monocyte populations can differentiate into macrophages or dendritic cells and augment tissue macrophages, they do not replenish tissue macrophages [155]. On contrary to longstanding scientific belief, tissue macrophages originate from embryonic progenitor cells and not from circulatory monocytes [155]. Tissue macrophages are given different names indicating their resident tissue such as Kupffer cells in the liver, microglia in the brain tissue, Langerhans cell in the skin, or alveolar macrophages in the lungs [156]. The other prominent phagocytizing population, PMNs or simply neutrophils, usually not present in healthy tissue, are located in bone marrow as they survive only a few days once released into circulation [157]. Consequently, they are used clinically to characterize infection, as a rising leukocyte population in peripheral blood is a solid indicator for an ongoing immune reaction [158]. Dendritic cells, either classical dendritic cells or plasmacytoid dendritic cells, ingest pathogens mainly to produce antigens and present them to effector cells such as lymphocytes [159]. 

Phagocytizing cells have, irrespective of previously described receptors, certain receptors for recognizing pathogens and triggering their killing by ROS. Besides TLRs, some of these receptors are made up of the C-type lectin family like Dectin-1, which recognizes β-glucan of fungal cell walls [160]. A further receptor, the mannose receptor, recognizes mannose, fucose, and *N*-acetylglucosamine sugar residues on the surface of broad variety of pathogens such as *Candida albicans*, *Leishmania donovani*, *Mycobacterium tuberculosis, and Klebsiella pneumoniae* [161,162]. Moreover, scavenger receptors are an important family of receptors that recognize unopsonized bacterial and parasitical microorganisms, targeting anionic polymers and acylated low-density lipoproteins on the cell surface of microbes using collagen domains like SR-A, SR-AII, and MARCO (macrophage receptor with collagenous structure) [163]. As more prominent receptors complement receptors and Fc receptors are described [160]. The complement system battles the pathogen invasion in three steps: Opsonization of bacteria toward engulfment by complement carrying receptors, subsequently, the inflammatory cells are recruited and the pathogens are killed by bacterial membrane perforation [164]. In Fc-mediated phagocytosis, antibody-opsonized particles are initially recognized, causing clustering of Fc receptors and subsequent reorganization of actin-associated proteins and formation of a phagocytic cup [165]. Another important step in pathogen-defense is enhancing gene expression of iNOS, thus enabling the production of ROS to kill pathogens [166]. This is a process where oxygen is rapidly depleted in the cell to create a so-called respiratory burst of oxygen radicals. Here, the key protein complex is the NAPDH oxidase or phagocyte oxidase, which is membrane bound and not yet fully assembled in resting macrophages or neutrophils. The respiratory burst is run by a protein complex composed of several subunits and requires specific signaling for activation [157,167,168]. The fully-assembled complex uses NADPH and O_2_ to create superoxide anion (O_2_^–^), which in turn is translated to hydrogen peroxide by the superoxide dismutase [169]. Under the influence of the enzyme myeloperoxidase (MPO), H_2_O_2_ and chloride or iodide ions are converted to hypochlorite ions, for instance [170]. This already impressive arsenal of ROS is complementary to antimicrobial peptides like defensin, lactoferrin, or lysozyme [171]. Despite its potent protective ability, an exaggerated respiratory burst of host cells, which are required for the process of killing microbes, can cause extensive tissue damage.

Impairment in phagocytes and ROS production, hereof neutrophils and macrophages, by alcohol, remains a distinctive problem in case of both acute and chronic alcohol consumption. Mice given alcohol resulting in blood levels of 100 to 120 mg/dl were exposed to a full-thickness excision [172]. The wounds, which were collected at 12 and 24 h after injury, show that MPO in wounds from the alcohol group is reduced compared to controls. Importantly, the histological assessment of the lesions indicates a significant difference in neutrophil infiltration [172]. To further investigate the relevance of MPO, three strains of mice with hyporesponsive TLR4, MyD88, and MPO knockout were exposed to alcohol (4 or 6 g/kg, 32% v/V) and injected with *E. coli* (1.5 × 10⁸/mouse) [173]. Mice with hyporesponsive TLR4 survive better than those with normal TLR4, while the lack of MyD88 or MPO does not markedly alter survival in the presence or absence of alcohol. Nonetheless, alcohol decreases the clearance of bacteria by macrophages and neutrophils in the peritoneal cavity and survival as well. Thus, the authors show that finding that wild-type and MPO knockout mice are similarly and significantly susceptible to alcohol-induced mortality in sepsis indicates that alcohol does not act primarily by inhibiting expression or function of MPO [173]. Nonetheless, adjacent to MPO, there are other proteins involved in the production of ROS as well, thus making the general production of ROS a functional assay for examination. Rats fed with an alcohol containing a Lieber-DeCarli diet (36% of calories) for seven days were sacrificed after eight days, and PMNs were isolated [174]. After that pre-opsonized pathogens were added (bacteria:cell ratio of 10:1) and intracellular production of ROS was assessed [174]. Chronic alcohol consumption reduces the production of ROS, as shown by the reduced ROS release by PMNs [174]. Another publication from the same lab investigated the role of acute alcohol exposure. Rats were injected with a bolus of alcohol (1.75 g/kg) and then infused with alcohol-containing liquids (250 mg/kg/h) for three hours [175]. Three hours after injection, N-formyl-methionyl-leucyl-phenylalanine (fMLP)-induced chemotaxis and superoxide release are significantly increased (2–3-fold) [175]. This correlates nicely with other data showing alcohol-reduced ingestion and intracellular killing by PMNs in a dose-dependent manner (8.5–85 mM), although ROS production is evidently increased, and there is no notable effect on phagocytic activity [175]. On the other hand, the capacity of neutrophils to phagocyte virulent *K. pneumoniae* is proportionally reduced by the presence of alcohol according to its concentration up to 800 mM [176].

In another in vivo model of ischemia/reperfusion of the brain, as performed via occlusion of both carotids, a single bolus of alcohol (blood levels around 42–46 mg/dl) and administration of apocynin, a specific inhibitor of NADPH oxidase before alcohol exposure, increases the translocation of subunit p67 from cytosol to membrane, and enhances NADPH oxidase function [177]. Similar data are observed in the Nagy group showing that ROS production greatly increases (2.5-fold) with the levels of Rac1-GTPase activity and the p67 translocation [178]. In mice which are treated with either alcohol (5 g/kg/day for 10 days) or poly I:C (a TLR4 agonist), or both, enhanced microglia activity is found in the combined insult group in histological analyses [179]. Even more, levels of NADPH oxidase gp91^phox^ mRNA are enhanced as well [179]. This is supported by another work with a chronic alcohol consumption model showing enhanced microglia activation in brain tissue [180]. Underlining this, alcohol treatment induces mitochondrial ROS production in cultured microglia [180]. There are also post-mortem analyses of brain tissue from patients with AUD that reveal significantly more tissue positive for microglia markers Iba-1 and GluT_5_ [181].

Alveolar macrophages are prone to oxidative stress as induced by alcohol as well. Indeed, several in vivo models of chronic alcohol consumption show abolished levels of glutathione in the BAL upon exposure, an effect that is associated with impaired alveolar macrophage function [182,183]. Yet again, upregulation of NADPH oxidase by chronic alcohol is identified as the pivotal factor in the impairment of the cellular function of alveolar macrophages in this model [183]. Yeligar et al. also provide evidence that chronic alcohol consumption depletes PPARγ reserves in alveolar macrophages, and that the addition of PPARγ ligands can re-establish their phagocytic ability [184]. A more recent work to further investigate the sensitization of Kupffer cells by alcohol using the aforementioned model reveals an intriguing role of microRNA [185]. More precisely, miRNA 181b-3p, that mediates importin 5α expression and sensitivity of TLR4 signaling, highlights a much more complex cell modulation by alcohol as assumed before [185]. Kupffer cells isolated from rats after single injection of alcohol (4 g/kg) exhibit tolerance to LPS at two hours after alcohol intake but are sensitized to LPS 24 hours later, exemplified by reduced TNFα release and NO production. This highlights a biphasic modulation by alcohol [186,187].

Additionally, disregarding the specificity of the innate immunity, the influence of alcohol-induced oxidative stress on cardiovascular system has to be considered as well. Rats subjected to chronic alcohol consumption (4 g/kg/day for 12 weeks) exhibit a significant increase in blood pressure compared with controls [188]. In addition, NADPH oxidase activity, membrane, and cytosolic p22^phox^ and p47^phox^ protein expression are elevated as well in the aortic tissue [188].

Summarizing this, it is evident that alcohol significantly impacts different cells of the innate immune arm, and different tissues, by modulating phagocytosis and/or oxidative burst. Mostly chronic alcohol consumption goes hand in hand with impaired macrophage and/or neutrophil functions.

### 6.2. Cellular Responses—Leukocyte Recruitment and Extravasation

Neutrophils migrate to their target tissue in a process termed extravasation, which is orchestrated by adhesion molecules and involves endothelial cells [189]. The first stage of this process is to roll along the endothelium in order to get to the aimed location. In order to do so, neutrophil surface proteins like sulfated Sialyl-Lewis X stick to endothelial membrane-bound proteins of the selectin family. For instance, glycoproteins with a lectin-like domain, namely P-selectin and E-selectin [190]. They are; however, only found in significant amounts after activation of endothelium by leukotrienes, complement, LPS, or TNFα, for example. These protein-to-protein interactions are reversible and not particularly strong, thus leading to bonds being released and formed along the lumen of a vessel [191]. The presence of chemokines such as CXCL8 to its respective receptor on the leukocyte enables activation of integrin receptors that induce conformational changes to enhance adhesion [123]. Nomenclature, as often seen in immunology, can be misleading because said integrin is called leukocyte functional antigen 1 (LFA-1) for historical reasons. These receptors, composed of two subunits, are bi-directional receptors because they transfer signals into and out of the cells [192]. In mammals, there are eighteen alpha subunits and eight beta subunits, with LFA1 having an αL/β2 or CD11a:CD18 heterodimer, while other rather important integrins are CR3 or MAC-1(CD11b:CD18), also a receptor for the C3b complement and factor X [193]. The third integrin of significance in this process is VLA4 (very late antigen or α4/β1 integrin, also called very late antigen 4 (VLA4), which binds to endothelial). Endothelial cells express the ligands for leukocytic integrins, such as ICAMs of the immunoglobulin-like superfamily, which are membrane bound [194]. In homeostasis, regular protein expression of VCAMs and ICAM2 is mostly sufficient for monocytic transmigration and adhesion to tissues [195]. Upon inflammation induction of ICAM-1 and upregulation of VCAMs and ICAM-2, the junction between leukocyte and vessel cells become enhanced [196]. Subsequently, LFA-1 bind to ICAM-1 and ICAM-2 resulting in a leukocyte arrest [196]. Transmigration through the endothelium is the next step for the leukocyte, as it forms pseudopods between endothelial cells and squeezes through them using other adhesion molecules like the platelet/endothelial–cell adhesion molecule (PECAM) or CD31 and junctional adhesion molecules (JAMs) or CD322 [197]. Interestingly, PECAM is presented by both, leukocytes and endothelial cells, interacting with each other [198]. The final barrier is the basement membrane, which has to be penetrated with matrix metalloproteinases like MMP-9 [199]. The leukocytes follow the chemotactic signal of tissue cells at the site of inflammation, which secrete chemokines that build up a gradient and lead the leukocyte to their target.

The issue of leukocyte migration in the presence of alcohol as well as pathogens is a common sight every day in clinical practice. In a prospective clinical study of precariously ill non-trauma patients, those individuals who were acutely intoxicated with alcohol have markedly diminished quantities of CRP, circulating neutrophils, and neutrophil CD64 indices [200]. The capability of neutrophils to roam blood vessels in order to find inflammation spots is not only crucial to the innate immune system but also visible in the expression of aforementioned adhesion molecules [196]. In a carrageenan air pouch model of mice subjected to bolus injection of alcohol (1.5 g/kg) and with LPS (1 ug/mL) afterward, the expression of adhesion molecules was investigated [201]. Alcohol inhibits TNF-mediated cell activation significantly and reduces leukocyte recruitment up to 90%. More distinctively, adhesion molecules ICAM-1, VCAM1, and E-selectin, as well as chemokines like CXCL8, MCP-1, and RANTES (“Regulated And Normal T cell Expressed and Secreted”, also known as CCL-5) are significantly reduced [201]. In another model of acute alcohol exposure, injection of 5.5 g/kg alcohol intraperitoneally significantly prevents the *E. coli* endotoxin-induced (112.5 ug/rat) expression of CD11b/c and CD18 on PMNs [202]. Interestingly, similar to the above-described differential impact of either acute or chronic alcohol consumption, in a chronic binge drinking model of mice, hepatic E-selectin expression enhances ten-fold, although expression of other adhesion molecules, including P-selectin, ICAM-1, and VCAM1, remains unaffected by alcohol [203]. The authors suggest that E-selectin may play an important role in neutrophil migration [203]. Furthermore, another chronic alcohol consumption model underlines a decrease of PMNs chemotaxis after LPS stimulation in alcohol-fed mice [204]. A further publication shows that alcohol may not only affect the general chemotaxis and migratory behavior of PMNs, but can modulate different steps of neutrophil infiltration in even contrasting directions as well [205]. Here, an in vitro model of alcohol (0.3% by vol.) exposure indicates no effect on PSGL-1, L-selectin, or CD11b expression, but does show altered distribution of PSGL-1 by alcohol. Additionally, alcohol prevents fMLP-mediated upregulation of CD11b and adhesion efficacy and increases membrane tether length and membrane growth up to three times [205]. Interestingly, the rolling velocity is also reduced by 55% compared to control cells [205]. In male Sprague-Dawley rats which have been nourished with Sustacal liquid diet (Mead Johnson, Evansville, IN) supplemented with 36% alcohol for four months, PMNs isolated from blood samples have upregulated CD18 expression on neutrophils to double that of control rats [206].

Alcohol influences the hematopoiesis of leukocytes by the modulation of adhesion molecules as well [207]. In an acute alcohol consumption mice model with subsequent intrapulmonary infection with *S. pneumoniae*, the infection-induced neutrophil recruitment to lung tissue is impaired [207]. Here again, this finding is associated with decreased bacterial clearance and higher mortality in the alcohol group [207]. Interestingly, analysis of bone marrow of the rodents reveals an inhibition of granulocyte precursor expansion, which would be a physiological response to infection [207]. In addition, chronic alcohol consumption plus acute alcohol intoxication suppresses the increase in blood granulocyte counts following intrapulmonary challenge with *S. pneumoniae* [208]. This suppression is associated with a significant decrease in bone marrow granulopoietic progenitor cell proliferation [208].

In conclusion, alcohol influences the various components of the innate immunity in different directions depending on its dose and the duration of exposure. Summarized, despite numerous studies, yet little is known about its interactions on the human body. For now, we have to acknowledge, due to the lack of knowledge, that Homer Simpson may have been right.

## 7. Structural Responses

Beside the immune cells-mediated host defense, mucous epithelial cells provide a physical barrier and contribute to regulation of innate and as well adaptive immunity. In the last years, microbiota has been extensively studied regarding its impact on various diseases. There is also evidence that alcohol abuse disrupts those epithelial barriers in gastrointestinal and respiratory tracts. Because this review focuses on the alcohol-mediated innate immune response, we discuss this topic only briefly.

Following chronical excessive alcohol intake, the intestinal barrier becomes “leaky” by altered tight junctions of epithelial cells. Here, more mechanisms have been described. On the one hand, alcohol impairs the trafficking of zona occludens (ZO)-1 and occludin, both proteins of tight junctions [209]. On the other hand, patients with alcoholic liver disease display increased intestinal levels of miR-212 that in turn binds the ZO-1 mRNA and impedes its synthesis [210]. The subsequent increased gut permeability enables the translocation of viable bacteria and their metabolites, toxins, and further DAMPs and PAMPs from intraintestinal lumen into extrainestinal space, reaching the liver by circulation, where it contributes to development of alcoholic liver disease [211,212]. Interestingly, chronic alcohol abuse causes leaky gut-dependent malabsorption in the small intestine that is comparable with untreated celiac disease [213]. Further, despite the increased intestinal permeability, bacterial overgrowth and compositional disbalance has been described. Patients with chronic alcohol overconsumption show lowered counts of protective gastrointestinal bacteria such as *Lactobacillus*, *Faecalibacterium,* or Bacteroidetes, whereby the pathogenic bacterial families such as Proteobacteria, Enterobacteriaceae, and Streptococcaceae were overrepresented [214]. The disbalance of intestinal bacterial composition as well the disruption of epithelial integrity seems to not be affected by a single alcohol binge, suggesting that the saying “the dose makes the poison” is correct [215].

Similarly to the intestine, the lung epithelial barrier is affected by chronical alcohol abuse as well, contributing to the pathophysiology of acute respiratory distress syndrome or acute lung injury. The interference of the granulocyte/macrophage colony stimulating factor signaling inhibits the macrophage maturation needed for maintenance of epithelial barrier integrity [216], and the bronchial epithelium ciliary function required for mechanical bacteria clearance has also been reported to be impaired [217]. Both lead to higher vulnerability to lung infections.

## 8. Limitations

This review, briefly summarized in Table 1, covers a broad aspect of innate immunity modulated by alcohol. However, further important aspects have not been included and have to be examined. For instance, dendritic cells and natural killer cells, as important parts in direct cell-mediated resistance to pathogens and other stimuli, have to be addressed in the context of alcohol’s immuno-modulatory properties. This has been reviewed in part by the Szabo group as well as the Kovacs group [218,219]. A great number of studies have also focused on the induction of apoptosis in PMNs and their longevity in the case of inflammation. This review did not address this issue and did not include those studies. Modern research has also proven that innate and adaptive immunity are not two separate compartments of immunity but, rather, interchanging and simultaneous components of host defense. Therefore, it is of crucial interest to look into adaptive immunity when discussing the effect of alcohol on the human body. A work by Pasala et al. is recommended here [220]. With a few exceptions, our understanding of the influence of alcohol on healthy individuals is rather limited and modern assays, as well as studies with this population, might offer new perspectives into alcohol research.

## 9. Conclusions

This review provides only a few of the potential mechanisms by which alcohol modulates organ injury and immunity. Although alcohol metabolism explains the inflammation-promoting impact of alcohol, there is no clear pathomechanistical explanation for its immune-suppressing effects. It is clear that chronic alcohol ingestion constitutes a permanent stress to the organism, with exacerbated organ response through the gut–liver–lung pro-inflammatory reactions. Here, alcohol-induced oxidative stress, impaired capacity of, for example, alveolar macrophages to phagocytize and/or clear bacteria, neutrophil sequestration, etc., are of the highest relevance for the clinical scenario, since those mechanisms increase the severity of a patient’s illness. Thus, the multifactorial and complex effects of alcohol on multiple-organ systems are closely associated with clear negative clinical outcomes, which also have been reproduced in in vivo and in vitro studies. The experimental data confirm the importance of diverse treatment strategies to reverse those effects by inhibiting the signal transduction, and to improve organ integrity as well as outcomes. Disturbances of immune function upon chronic alcohol use on a variety of diverse conditions, such as liver disease, lung disease, cancer, traumatic injury, but also bacterial and viral infections, are proven. However, as this review has already indicated, alcohol exposure also has profound anti-inflammatory effects depending on its use. Although newer studies uncover the role of an inflammatory “switch” upon acute alcohol use, as briefly noted in this review, particularly the impact of acute alcohol ingestion on innate but also on adaptive immunity is poorly studied. Of course, the immune-suppressive effects may hide some benefits, as shown in models and also clinical studies, in cases of traumatic injury. The provided explanations are evident, since the traumatic insult itself induces a strong inflammatory response, which may harm the patient’s body. However, reducing this potentially overwhelming response may protect from some inflammation-related organ injuries in an acute setting. Most of the experimental studies confirm the anti-inflammatory effects on the one hand; however, they raise important concerns about the “prolonged” immune-suppressive influence of acute alcohol ingestion, which may increase, for example, the risk for infections on the other hand. Thus, alcohol-related impairments of the innate immune responses, that have been addressed here, may contribute to accelerated disease progress. Although the signal sensing and its transduction under conditions of alcohol intoxication have been studied in vivo and in vitro, the molecular mechanisms are still not fully understood. Neither are there studies considering the cross-talk of innate and adaptive immune responses under acute alcohol intoxication. Furthermore, little is known on the effects of alcohol-mediated innate immunity with regard to ageing or gender issues. Thus, more work is necessary to examine whether there are certain critical effects which should be considered in the clinical scenario in elderly patients. In our own study, we aimed at investigating the mechanism underlying influence of alcohol on specific NF-ĸB signaling during the inflammatory response of human lung epithelial cells [221]. Stimulating cells with either IL-1β or sera from trauma patients induced a strong release of IL-6 as well as increased neutrophil adhesion to epithelial cells. Consecutive treatment with alcohol markedly decreased both IL-6 release and neutrophil adherence. Regarding the specific NF-ĸB signaling, IL-1β induced a significant activation of both the canonical (p50) and non-canonical (p65) pathways. Stratification of impact of alcohol has shown that alcohol was able to significantly reduce p50, representing the canonical pathway, but not p52 activation, representing the non-canonical pathway of NF-ĸB signaling. As an example, this mechanistical study provides some basis for further research considering the opposite effects of acute versus chronic alcohol use, in order to design therapeutics targets for immune alterations in individuals with various drinking behaviors. 

## Figures and Tables

**Figure 1 jcm-08-01981-f001:**
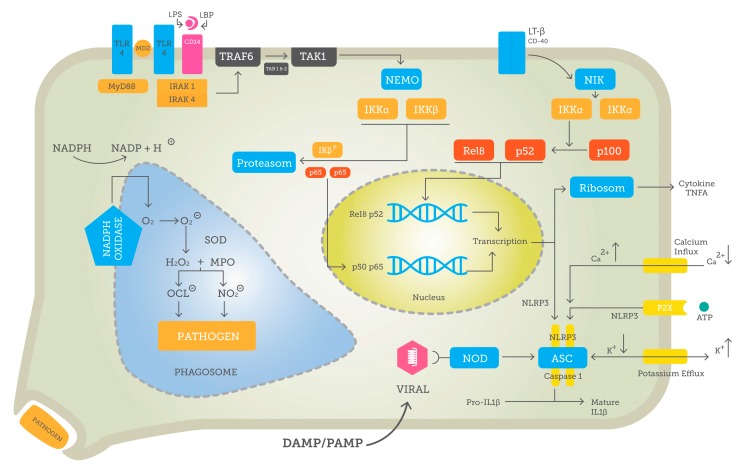
Potential intracellular target points for (i) acute alcohol and (ii) chronic alcohol in a stylized cell. The induction of canonical NF-κB with p50–p65 translocation to nucleus via pattern recognition receptors (PRR) is outlined by, for example, TLR4 and MyD88 activation. The non-canonical NF-κB pathway with p52-RelB is detailed with CD40 as the respective receptor. Either pathway leads to the transcription of inflammatory cytokines (e.g., TNF-α or important immune regulatory proteins potentiating, for example, inflammasome formation.) Inflammasome formation itself is comprised of ASC, Caspase-1, and NLRP3 and regulated via ionic currents or intracellular PRRs, like nucleotide-binding oligomerization domain (NOD). Another cell compartment under alcohol influence is the phagosome needed for ingestion and destruction of pathogens using an array of reactive oxygen species.

**Figure 2 jcm-08-01981-f002:**
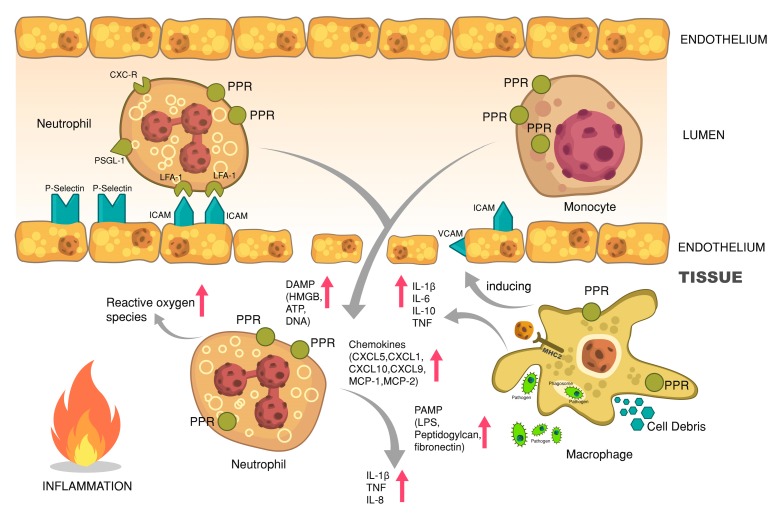
Potential target points for (i) acute alcohol and (ii) chronic alcohol in inflammatory tissue. Neutrophils and monocytes migrate towards the site of inflammation via adhesion molecules like P-selectin and intercellular adhesion molecule (ICAM), which can be induced by activated macrophages. This process of transmigration is further enhanced by secretion of chemokines and PAMPs as well as DAMPs. The process of phagocytosis is shown as an example in macrophages that clear the tissue of pathogens and cell debris. Neutrophils also produce reactive oxygen species that damage pathogens, as well as healthy tissue, and enhance endothelium leakiness.

**Table 1 jcm-08-01981-t001:** Brief overview of some factors involved in alcohol-modulated immunity.

Alcohol Intake	Pattern Recognition				
	Species	Study	Cell Type/Organ	Major Finding	Reference
acute	human	in vitro	monocytes	↓ TLR8- and TLR4-mediated TNFα synthesis (mRNA, protein)	[26,27]
		in vitro	monocytes	↓ TLR4-dependent NFκB activation	[27]
	mouse	in vivo	liver	↓ TLR4 gene expression	[21]
		in vivo	intestine	↓ TLR4-mediated cytokine production (sepsis model)	[23]
		in vitro	microglial cells	↑ TLR4 and TLR2 mRNA at 1 h ↓ TLR4 and TLR2 mRNA at 3 h	[24]
		in vivo	macrophages	↓ TLR2, TLR4 and TLR9 induced inflammatory response mediated by p38 and ERK1/2	[25]
	rat	in vivo	intestine	↓ TLR4 sensitivity to ligands (LPS)	[22]
chronic	mouse	in vivo	liver macrophages	↑ TLR4 protein expression	[29]
		in vivo	liver	↑ all TLR mRNAs except TLR3 and TLR5 (TLR10 and TLR11 not tested) ↓ antibiotics effectiveness toward TLR mRNA expression ↓ TNFα protein level by NADPH oxidase inhibition	[29]
	NF-κB signaling				
acute	human	in vitro	monocytes	↑ p50 translocation to the nucleus ↓ cytokine levels = p65/RelA	[54,55]
		in vitro	FEMX-I melanoma cells	↑ NF-κB activation	[56]
		in vitro	monocytes	↓ NF-κB activity independently on IκB α degradation	[57]
		in vitro	monocytes	↑ IRAK-M ↓ IκB α kinase activity, NF-κB DNA binding, and NF-κB driven cellular responses	[63]
	mouse	in vitro	alveolar type 2 epithelial cells	↓ CXCL5 expression ↓ p65 phosphorylation (following LPS and TNFα exposure)	[60]
		in vivo	kupffer cells	↓ IRAK-1 and LPS-mediated activation of NF-κB at 1 h in ↑ IRAK-1 and LPS-induced NF-κB activity at 24 h	[64]
	rat	in vitro	kupffer cells	↓ LPS-induced NF-κB activation ↓ TNFα mRNA and TNFα secretion	[53]
		in vivo	peritoneal macrophages	↓ NF-κB activation = LPS-induced p65 translocation	[59]
chronic	human	in vitro	monocytes	↓IRAK-M activity ↑IRAK-1 and IKK kinase levels, NF-κB DNA binding and responses ↑ ERK activation	[63]
	mouse	in vivo	kupffer cells, leukocytes, liver	↑NF-κB phosphorylation ↑ NF-κB activation	[61,62]
	Bacterial clearance				
acute	human	healthy volunteers	blood	↑ serum endotoxin ↑ 16 S rDNA levels	[83]
	mouse	in vivo	peritoneal cavity + spleen; intestinal barrier, blood	↓ bacterial clearance ↑ overgrowth of bacteria, and ↑ translocation of pathogens into the bloodstream	[23,90]
	rat	in vivo	bacteria numbers in the lungs, blood	delayed lung neutrophil recruitment ↑ elevated bacterial burden, systemic dissemination to spleen, blood, and liver	[81]
chronic	human			↑ bacterial infections	[85]
	rats	in vivo	systemic, gut mucosa	↑ endotoxemia	[87]
	Cytokines				
acute	human	in vitro	lung epithelial cells	↓ IL-8 and IL-6 release	[127,128]
		in vitro	monocytes	↓ TNFα release and gene expression ↑ IL-10	[27,63]
		healthy volunteers	blood	↑ TNFα after 20 minutes, followed by ↓ IL-1β after 2-5 hours	[134]
		trauma patients	blood	↓ IL-6 ↑ TNFα after 20 minutes, followed by ↓ IL-1β after 2-5 hours	[137,138]
	mouse	in vivo	blood	↓ CXCL9, IL-6 and IL-12 ↑ IL-10	[129]
		in vivo	peritoneal lavage fluid	↓ IL-15, TNFα, IL-9, IL-1β & IL-1α, IL-13, IL-17, and IL-6 ↑ IL-10 and MIP-2	[23]
		in vivo	blood, lung	↑ IL-6 and TNFα ↑ IL-6	[139,140]
		in vivo	ileal tissue	↑ IL-6, IL-1β and IL-18 protein	[141,142]
chronic	human	in vitro	monocytes	↑ TNFα = IL-10 protein level	[28,63]
		alcoholic liver disease	blood	↑ TNFα, IL-6 and IL-1	[131]
		acute alcoholic hepatitis	blood	↑ IL-8, IL-4, and IFNγ	[132]
	mouse	in vivo	liver	↑ mRNA expression of TNFα, IL-6 and IL-10 ↑ TNFα level	[29,130]
	rat	in vivo	blood	↑ IL-1, IL-10 and TNFα	[147]
		in vivo	blood	= TNFα, IL-10, IL-6 or CXCL1 without inflammatory trigger, but ↑of IL-6 and TNF upon stimulation	[148]
		in vivo	blood	↑ IL-6, MCP-1 and TNFα	[61]
	Cellular responses				
acute	human	precarious ill patients	blood	↓ CRP, circulating neutrophils and neutrophilic CD64	[200]
	mouse	in vivo	wounds	↓ MPO and neutrophilic infiltration	[172]
		in vivo	leukocytes	↓ cellular recruitment ↓ adhesion molecules ICAM-1, VCAM1, and E-selectin ↓ chemokines like CXCL8, MCP-1 and RANTES	[201]
		in vivo	lung neutrophils	↓ recruitment	[207]
	rat	in vivo	isolated PMN	↑ chemotaxis and superoxide release ↓ ingestion and intracellular killing	[175]
		in vivo	isolated PMN	↓ reduced phagocytosis of virulent K. pneumoniae	[176]
chronic	mouse	in vivo	brain tissue, microglia	↑ microglia activation	[180]
		in vitro	microglia	↑ mitochondrial ROS production	[180]
		in vivo	alveolar macrophages	↓ cellular functions e.g. phagocytosis	[183,184]
		in vivo	liver	↑ E-selectin expression = expression of P-selectin, ICAM-1, and VCAM1	[184]
	rat	in vivo	isolated PMN	↓ ROS	[174]
		in vivo	systemic, gut mucosa	↑ ROS	[87]
		in vivo	liver	modulation of different steps of neutrophil infiltration	[186]
		in vivo	isolated PMN	↑ CD18 expression	[187]
	Others				
acute	human	in vitro	neuroblastoma cells	↑ HMGB1 expression and release	[91]
		in vitro	macrophages	↑ IL-1β and inflammasome activation	[145]
	mice	in vivo	brain	↑ P2X7R in alcohol sensitive brain regions	[92]

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
