# Peer review of "Innate Immunity and Alcohol"

_jcm, 2019, doi:10.3390/jcm8111981_

Round 1

Reviewer 1 Report

Journal of Clinical Medicine

Reviewer's report

Title: Innate Immunity and Alcohol

Date: 25 October 2019

General remarks:

The manuscript “Innate Immunity and alcohol” deals with interesting topic. The topic of the work closely related to scope of the journal. However, many aspects must be fulfilled:

the authors should provide information about the influence of alcohol’s consumption on certain factors of innate immunity on the levels of genes. One of the most talked about topic is gut microbiome and their effect on various diseases. The authors should touch this topic as to how alcohol shapes the gut microbiome and modulates the innate inflammatory function. The results presented in this manuscript should be included in table. the authors should provide more information about the influence of alcohol’s consumption on certain factors of innate immunity in the case of man.

Author Response

Dear reviewer,

that you very much  for your valuable comment!

We have revised our review according to your suggestions, and included as example a section on gut microbiome, a table with a literature overview etc. Furthermore, we have included a brief discussion on the topic.

We hope, that we were able to improve our manuscript by the made revision.

Thank you

Reviewer 2 Report

Interesting review. But some chapters require to be better detailed

In the innate immunity chapter, the authors should more precisely describe the role of each TLR, and mention the PAMPs which bind to them. A scheme should be shown to summarize this chapter of particular importance, because the title adresses « Innate immunity and Alcohol ». In the chapter NFKb , » Traf2 and Traf3 bind to NFKB » requires a precision about the surface molecules that  activate the binding of these TRAF molecules to NFKb. In pathogen-associated sterile inflammation, the authors should introduce the different inflammasomes, and describe the role of caspase 1. Moreover, the authors should more clearly indicate the function of purinergic receptors. In the chapter cytokines , are missing the role of IL-1b in Diabetes, as discovered by Dinarello himself, the role of TNFa in tuberculosis, and IL-10 in Tregs. IL-6 is missing In the chapter on cellular responses, among the surface molecules expressed on phagocytozing cells, and recognizing pathogens are TLRs. This has been omitted. The sentence in page 11, lane 7, should begin by « besides TLRs, some of these receptors are made up of lectin… In the chapter on leukocyte recruitment, the corresponding receptors  of Icam1,  Icam2 Vcam Pcam should be mentioned. There is no perspectives in the discussion. This is missing In most of the chapters, data reporting alcohol-mediated uprugulation of molecule expression or signalisation should be gathered and opposed with  those reporting a decrease. A short discussion should be adressed to try  to explain those discordances, if this is known.

Author Response

Dear reviewer,

thank you very much for your valuable comments! 

We have revised our manuscript according to your suggestions, and all changes can be found in the newly submitted version.

We have addressed all issues that were raised, and if some additional work is required, we are happy to provide further revisions.

Thany you